# A STRONG ON-POLICY COMPETITOR TO PPO

## ABSTRACT

As a recognized variant and improvement for Trust Region Policy Optimization (TRPO), proximal policy optimization (PPO) has been widely used with several advantages: efficient data utilization, easy implementation, and good parallelism. In this paper, a first-order gradient reinforcement learning algorithm called Policy Optimization with Penalized Point Probability Distance (POP3D), which is a lower bound to the square of total variance divergence, is proposed as another powerful variant. The penalty item has dual effects, prohibiting policy updates from overshooting and encouraging more explorations. By carefully controlled experiments on both discrete and continuous benchmarks, our approach is proved highly competitive to PPO.

## 1  INTRODUCTION

With the development of deep reinforcement learning, lots of impressive results have been produced in a wide range of fields such as playing Atari game (Mnih et al., 2015; Hessel et al., 2018), controlling robotics (Lillicrap et al., 2015), Go (Silver et al., 2017), neural architecture search (Tan et al., 2019; Pham et al., 2018).

The basis of a reinforcement learning algorithm is generalized policy iteration (Sutton & Barto, 2018), which states two essential iterative steps: policy evaluation and improvement. Among various algorithms, policy gradient is an active branch of reinforcement learning whose foundations are *Policy Gradient Theorem* and the most classical algorithm REINFORCEMENT (Sutton & Barto, 2018). Since then, handfuls of policy gradient variants have been proposed, such as Deep Deterministic Policy Gradient (DDPG) (Lillicrap et al., 2015), Asynchronous Advantage Actor-Critic (A3C) (Mnih et al., 2016), Actor-Critic using Kronecker-factored Trust Region (ACKTR) (Wu et al., 2017), and Proximal Policy Optimization (PPO) (Schulman et al., 2017).

Improving the strategy monotonically had been nontrivial until Schulman et al. (2015) proposed Trust Region Policy Optimization (TRPO), in which Fisher vector product is utilized to cut down the computing burden. Specifically, Kullback–Leibler divergence (**KLD**) acts as a hard constraint in place of objective, because its corresponding coefficient is difficult to set for different problems. However, TRPO still has several drawbacks: too complicated, inefficient data usage. Quite a lot of efforts have been devoted to improving TRPO since then and the most commonly used one is PPO.

PPO can be regarded as a first-order variant of TRPO and have obvious improvements in several facets. In particular, a pessimistic clipped surrogate objective is proposed where TRPO's hard constraint is replaced by the clipped action probability ratio. In such a way, it constructs an unconstrained optimization problem so that any first-order stochastic gradient optimizer can be directly applied. Besides, it's easier to be implemented and more robust against various problems, achieving an impressive result on Atari games (Brockman et al., 2016). However, the cost of data sampling is not always cheap. Haarnoja et al. (2018) design an off-policy algorithm called Soft Actor-Critic and achieves the state of the art result by encouraging better exploration using maximum entropy.

In this paper, we focus on the on-policy improvement to improve PPO and answer the question: how to successfully leverage penalized optimization to solve the constrained one which is formulated by Schulman et al. (2015).

1. It proposes a simple variant of TRPO called POP3D along with a new surrogate objective containing a point probability penalty item, which is symmetric lower bound to the square of the total variance divergence of policy distributions. Specifically, it helps to stabilize

the learning process and encourage exploration. Furthermore, it escapes from penalty item setting headache along with penalized version TRPO, where is arduous to select one fixed value for various environments.

2. It achieves state-of-the-art results among on-policy algorithms with a clear margin on 49 Atari games within 40 million frame steps based on two shared metrics. Moreover, it also achieves competitive results compared with PPO in the continuous domain. It dives into the mechanism of PPO's improvement over TRPO from the perspective of solution manifold, which also plays an important role in our method.

3. It enjoys almost all PPO's advantages such as easy implementation, fast learning ability.

We provide the code and training logs to make our work reproducible.

## 2 PRELIMINARY KNOWLEDGE AND RELATED WORK

### 2.1 POLICY GRADIENT

Agents interact with the environment and receive rewards which are used to adjust their policy in turn. At state $s_t$, one agent takes strategy $\pi$ and transfers to a new state $s_{t+1}$, rewarded $r_t$ by the environment. Maximizing discounted return (accumulated rewards) $R_t$ is its objective. In particular, given a policy $\pi$, $R_t$ is defined as

$$R_t = \sum_{n=0}^{\infty} (r_t + \gamma r_{t+1} + \gamma^2 r_{t+2} + ... + \gamma^n r_{t+n}). \tag{1}$$

$\gamma$ is the discounted coefficient to control future rewards, which lies in the range $(0, 1)$. Regarding a neural network with parameter $\theta$, the policy $\pi_\theta(a|s)$ can be learned by maximizing Equation 1 using the back-propagation algorithm. Particularly, given $Q(s, a)$ which represents the agent's return in state $s$ after taking action $a$, the objective function can be written as

$$\max_\theta \quad \mathbb{E}_{s,a} \log \pi_\theta(a|s) Q(s, a). \tag{2}$$

Equation 2 lays the foundation for handfuls of policy gradient based algorithms. Another variant can be deduced by using

$$A(s, a) = Q(s, a) - V(s) \tag{3}$$

to replace $Q(s, a)$ in Equation 2 equivalently, $V(s)$ can be any function so long as $V$ depends on $s$ but not $a$. In most cases, state value function is used for $V$, which not only helps to reduce variations but has clear physical meaning. Formally, it can be written as

$$\max_\theta \quad \mathbb{E}_{s,a} \log \pi_\theta(a|s) A(s, a). \tag{4}$$

### 2.2 ADVANTAGE ESTIMATE

A commonly used method for advantage calculation is one-step estimation, which follows

$$A(s_t, a_t) = Q(s_t, a_t) - V(s_t) = r_t + \gamma V(s_{t+1}) - V(s_t). \tag{5}$$

However, a more accurate method called generalized advantage estimation is proposed in Schulman et al. (2016), where all time steps of estimation are combined and summarized using $\lambda$-based weights,. The generalized advantage estimator $\hat{A}_t^{GAE(\gamma,\lambda)}$ is defined by Schulman et al. (2016) as

$$\hat{A}_t^{GAE(\gamma,\lambda)} := (1 - \lambda) * (\hat{A}_t^{(1)} + \lambda \hat{A}_t^{(2)} + \lambda^2 \hat{A}_t^{(3)} + \dots) = \sum_{l=0}^{\infty} (\gamma\lambda)^l \delta_{t+l}^V$$

$$\delta_{t+l}^V = r_{t+l} + \gamma V(s_{t+l+1}) - V(s_{t+l}). \tag{6}$$

$$\hat{A}_t^{(k)} := \sum_{l=0}^{k-1} \gamma^l \delta_{t+l}^V = -V(s_t) + r_t + \gamma r_{t+1} + \dots + \gamma^{k-1} r_{t+k-1} + \gamma^k V(s_{t+k})$$

The parameter $\lambda$ meets $0 \leq \lambda \leq 1$, which controls the trade-off between bias and variance. All methods in this paper utilize $\hat{A}_t^{GAE(\gamma,\lambda)}$ to estimate the advantage.

## 2.3 TRUST REGION POLICY OPTIMIZATION

Schulman et al. (2015) propose TRPO to update the policy monotonically. In particular, its mathematical form is

$$\max_\theta \quad \mathbb{E}_t[\frac{\pi_\theta(a_t|s_t)}{\pi_{\theta_{old}}(a_t|s_t)}\hat{A}_t] - C\mathbb{E}_t[KL[\pi_{\theta_{old}}(\cdot|s_t), \pi_\theta(\cdot|s_t)]]$$
$$\epsilon = \max_s E_{a\sim\pi_\theta(a|s)}[A_{\pi_{\theta_{old}}}(s,a)]) \tag{7}$$

where $C$ is the penalty coefficient, $C = \frac{2\epsilon\gamma}{(1-\gamma)^2}$.

In practice, the policy update steps would be too small if $C$ is valued as Equation 7. In fact, it's intractable to calculate $C$ beforehand since it requires traversing all states to reach the maximum. Moreover, inevitable bias and variance will be introduced by estimating the advantages of old policy while training. Instead, a surrogate objective is maximized based on the KLD constraint between the old and new policy, which can be written as below,

$$\max_\theta \quad \mathbb{E}_t[\frac{\pi_\theta(a_t|s_t)}{\pi_{\theta_{old}}(a_t|s_t)}\hat{A}_t]$$
$$s.t. \quad \mathbb{E}_t[KL[\pi_{\theta_{old}}(\cdot|s_t), \pi_\theta(\cdot|s_t)]] \leq \delta \tag{8}$$

where $\delta$ is the KLD upper limitation. In addition, the conjugate gradient algorithm is applied to solve Equation 8 more efficiently. Two major problems have yet to be addressed: one is its complexity even using the conjugate gradient approach, another is compatibility with architectures that involve noise or parameter sharing tricks (Schulman et al., 2017).

## 2.4 PROXIMAL POLICY OPTIMIZATION

To overcome the shortcomings of TRPO, PPO replaces the original constrained problem with a pessimistic clipped surrogate objective where KL constraint is implicitly imposed. The loss function can be written as

$$L^{CLIP}(\theta) = \mathbb{E}_t[\min(r_t(\theta)\hat{A}_t, clip(r_t(\theta), 1-\epsilon, 1+\epsilon)\hat{A}_t)]$$
$$r_t(\theta) = \frac{\pi_\theta(a_t|s_t)}{\pi_{\theta_{old}}(a_t|s_t)}, \tag{9}$$

where $\epsilon$ is a hyper-parameter to control the clipping ratio. Except for the clipped PPO version, KL penalty versions including fixed and adaptive KLD. Besides, their simulation results convince that clipped PPO performs best with an obvious margin across various domains.

## 3 POLICY OPTIMIZATION WITH PENALIZED POINT PROBABILITY DISTANCE

Before diving into the details of POP3D, we review some drawbacks of several methods, which partly motivate us.

### 3.1 DISADVANTAGES OF KULLBACK-LEIBLER DIVERGENCE

TRPO (Schulman et al., 2015) induced the following inequality[1],

$$\eta(\pi_\theta) \leq L_{\pi_{\theta_{old}}}(\pi_\theta) + \frac{2\epsilon\gamma}{(1-\gamma)^2}\alpha^2$$
$$\alpha = D_{TV}^{\max}(\pi_{\theta_{old}}, \pi_\theta) \tag{10}$$
$$D_{TV}^{\max}(\pi_{\theta_{old}}, \pi_\theta) = \max_s D_{TV}(\pi_{\theta_{old}}||\pi_\theta)$$

TRPO replaces the square of total variation divergence $D_{TV}^{max}(\pi_{\theta_{old}}, \pi_\theta)$ by $D_{KL}^{\max}(\pi_{\theta_{old}}, \pi_\theta) = \max_s D_{KL}(\pi_{\theta_{old}}||\pi_\theta)$.

---

[1]Note that $\eta$ means loss instead of return as the ICML version (Schulman et al., 2015).

Given a discrete distribution $p$ and $q$, their total variation divergence $D_{TV}(p||q)$ is defined as

$$D_{TV}(p||q) := \frac{1}{2} \sum_i |p_i - q_i| \qquad (11)$$

in TRPO (Schulman et al., 2015). Obviously, $D_{TV}$ is symmetric by definition, while KLD is asymmetric. Formally, given state $s$, KLD of $\pi_{\theta_{old}}(\cdot|s)$ for $\pi_\theta(\cdot|s)$ can be written as

$$D_{KL}(\pi_{\theta_{old}}(\cdot|s)||\pi_\theta(\cdot|s)) := \sum_a \pi_{\theta_{old}}(a|s) \ln \frac{\pi_{\theta_{old}}(a|s)}{\pi_\theta(a|s)}. \qquad (12)$$

Similarly, KLD in the continuous domain can be defined simply by replacing summation with integration. The consequence of KLD's asymmetry leads to a non-negligible difference of whether choose $D_{KL}(\pi_{\theta_{old}}||\pi_\theta)$ or $D_{KL}(\pi_\theta||\pi_{\theta_{old}})$. Sometimes, those two choices result in quite different solutions. Robert compared the forward and reverse KL on a distribution, one solution matches only one of the modes, and another covers both modes (Murphy, 2012). Therefore, KLD is not an ideal bound or approximation for the expected discounted cost.

## 3.2 DISCUSSION ABOUT PESSIMISTIC PROXIMAL POLICY

In fact, PPO is called pessimistic proximal policy optimization[2] in the meaning of its objective construction style. Without loss of generality, supposing $A_t > 0$ for given state $s_t$ and action $a_t$, and the optimal choice is $a_t^\star$. When $a_t = a_t^\star$, a good update policy is to increase the probability of action to a relatively high value $a_t^\star$ by adjusting $\theta$. However, the clipped item $clip(r_t(\theta), 1 - \epsilon, 1 + \epsilon)\hat{A}_t$ will fully contribute to the loss function by the minimum operation, which ignores further reward by zero gradients even though it's the optimal action. Other situation with $A_t < 0$ can be analyzed in the same manner.

However, if the pessimistic limitation is removed, PPO's performance decreases dramatically (Schulman et al., 2017), which is again confirmed by our preliminary experiments. In a word, the pessimistic mechanism plays a very critical role for PPO in that it has a relatively weak preference for a good action decision at a given state, which in turn affects its learning efficiency.

## 3.3 RESTRICTED SOLUTION MANIFOLD FOR EXACT DISTRIBUTION MATCHING

To be simple, we don't take the model identifiability issues along with deep neural network into account here because they don't affect the following discussion much (LeCun et al., 2015). Suppose $\pi_{\theta\star}$ is the optimal solution for a given environment, in most cases, more than one parameter set for $\theta$ can generate the ideal policy, especially when $\pi_{\theta\star}$ is learned by a deep neural network. In other words, the relationship between $\theta$ and $\pi_{\theta\star}$ is many to one. On the other hand, when agents interact with the environment using policy represented by neural networks, they prefer to takes the action with the highest probability. Although some strategies of enhancing exploration are applied, they don't affect the policy much in the meaning of expectation.

RL methods can help agents learn useful policies after fully interacting with the environment. Take Atari-Pong game for example, when an agent sees a Pong ball coming close to the right (state $s_1$), its optimal policy is moving the racket to the right position (for example, the "RIGHT" action) with a distribution $p_{\theta_1}^{s_1} = [0.05, 0.05, 0.1, 0.7, 0.05, 0.05]^3$. The probability of selecting "RIGHT" is a relatively high value such as 0.7. It's almost impossible to push it to be 1.0 exactly since it's produced by a softmax operation on several discrete actions. In fact, we hardly obtain the optimal solution accurately. Instead, our goal is to find a good enough policy. In this case, the policy of pushing $p(\text{RIGHT}|s_1)$ above a threshold is sufficient to be a good one. In other words, paying attention to the most critical actions is sufficient, and we don't care much the probability value of the other non-critical actions. For example, a good policy at $s_1$ is $[?,?, \geq 0.7, ?,?,?]$. Note that $\pi_\theta(a|s)$ is represented by a neural network parameterized using $\theta$ and a good policy for the whole game means

---

[2]The word "pessimistic" is used by the PPO paper.
[3]The action space is described as ['NOOP', 'FIRE', 'RIGHT', 'LEFT', 'RIGHTFIRE', 'LEFTFIRE'].

that the network can perform well across the whole state space. Focusing on those critical actions at each state[4] and ignoring non-critical ones can help the network learn better and more easily.

Using a penalty such as KLD cannot utilize this good property, because it involves all of the actions' probabilities. Moreover, it doesn't stop penalizing unless two distributions become exactly indifferent or the advantage item is large enough to compensate for the KLD cost. Therefore, even if $\theta$ outputs $\theta_{old}$ the same high probability for the right action, the penalization still exists. Suppose that two parameters for $\theta_1$: $\theta_2$ and $\theta_3$, where $p_{\theta_2}^{s_1}$ = [0.01, 0.15, 0.05, 0.7, 0.01, 0.08] and $p_{\theta_3}^{s_1}$ = [0.01, 0.01, 0.01, 0.7, 0.26, 0.01]. When the agent already chooses RIGHT at $S_1$, the loss item from a good penalized distance should be small. However, $D_{KL}(\pi_{\theta_1}(\cdot|s_1)||\pi_{\theta_2}(\cdot|s_1))$=0.15 and $D_{KL}(\pi_{\theta_1}(\cdot|s_1)||\pi_{\theta_3}(\cdot|s_1))$=0.39. However, it's not necessary to require the distribution of other actions ('NOOP', 'FIRE', 'LEFT', 'RIGHTFIRE', 'LEFTFIRE') of $p_{\theta_2}^{s_1}$ near to $p_{\theta_1}^{s_1}$. Instead, it's better to relax this requirement to enlarge the freedom degree of the network and focus on learning important actions. Doing this brings another advantage, the agent can explore more for non critical actions. From the perspective of the manifold, optimal parameters constitute a solution manifold. The KLD penalty will act until $\theta$ exactly locates in the solution if possible, akin to mapping a point onto a curve. Instead, if the agent concentrates only on critical actions like a human does, it's much easier to approach the manifold in a higher dimension. This is comparable to expanding the solution manifold by at least one dimension, e.g. from curves to surfaces or from surfaces to spheres.

## 3.4 EXPLORATION

One shared highlight in reinforcement learning is the balance between exploitation and exploration. For a policy-gradient algorithm, entropy is added in the total loss to encourage exploration in most cases. When included in the loss function, KLD penalizes the old and new policy probability mismatch for all possible actions as Equation 12 given a state $s$. This strict punishment for every action's probability mismatch, which discourages exploration.

## 3.5 POINT PROBABILITY DISTANCE

To overcome the above-mentioned shortcomings, we propose a surrogate objective with the point probability distance penalty, which is symmetric and more optimistic than PPO. In the discrete domain, when the agent takes action $a$, the point probability distance between $\pi_{\theta_{old}}(\cdot|s)$ and $\pi_\theta(\cdot|s)$ is defined by

$$D_{pp}^a(\pi_{\theta_{old}}(\cdot|s), \pi_\theta(\cdot|s)) = (\pi_{\theta_{old}}(a|s) - \pi_\theta(a|s))^2. \tag{13}$$

Attention should be paid to the penalty definition item, the distance is measured by the point probability, which emphasizes its mismatch for the **sampled** actions for a state. Unless it would lead to confusion, we omit $a$ for simplicity in the following sections. Undoubtedly, $D_{pp}$ is symmetric by definition. Furthermore, it can be proved that $D_{pp}$ is indeed a lower bound for the total variance divergence $D_{TV}$. As a special case, it can be easily proved that for binary distribution, $D_{TV}^2(p||q) = D_{pp}(p||q)$.

**Theorem 3.1.** *For two discrete probability distributions p and q with K values, then $D_{TV}^2(p||q) \geq D_{pp}^a(p||q)$ holds for any action a and $\mathbb{E}_a D_{pp}^a(p||q)$ is a lower bound for $D_{TV}^2(p||q)$.*

---

[4]Note that some states don't have critical action. Taking the Pong for example, when the ball is just shot back, the agent can choose any action.

*Proof.* Let $p_l = \alpha, q_l = \beta$ for the $l$-th action $a$, and suppose $a \geq b$ without loss of generalization. So,

$$
\begin{aligned}
D_{TV}^2(p||q) &= (\frac{1}{2}\sum_{i=1}^{K}|p_i - q_i|)^2 = (\frac{1}{2}\sum_{i=1,i\neq l}^{K}|p_i - q_i| + \frac{1}{2}|p_l - q_l|)^2 \\
&\geq (\frac{1}{2}|\sum_{i=1,i\neq l}^{K}p_i - q_i| + \frac{1}{2}(\alpha - \beta))^2 = (\frac{1}{2}|1 - \alpha - (1 - \beta)| + \frac{1}{2}(\alpha - \beta))^2 \\
&= (\frac{1}{2}(\alpha - \beta) + \frac{1}{2}(\alpha - \beta))^2 = D_{pp}^a(p||q) \\
\mathbb{E}_a D_{pp}^a(p||q) &= \sum_a p(a)D_{pp}^a(p||q) \leq \sum_a p(a)D_{TV}^2(p||q) = D_{TV}^2(p||q)
\end{aligned}
$$

□

Since $0 \leq \pi_\theta(a|s) \leq 1$ holds for discrete action space, $D_{pp}$ has a lower and upper boundary: $0 \leq D_{pp} \leq 1$. Moreover, $D_{pp}$ is less sensitive to action space dimension than KLD, which has a similar effect as PPO's clipped ratio to increase robustness and enhance stability. Equation 13 stays unchanged for the continuous domain, and the only difference is $\pi_\theta(a|s)$ represents point probability density instead of probability.

## 3.6 POP3D

After we have defined the point probability distance, we use a new surrogate objective $f_\theta$ for POP3D, which can be written as

$$
\max_\theta \quad \mathbb{E}_t[\frac{\pi_\theta(a_t|s_t)}{\pi_{\theta_{old}}(a_t|s_t)}\hat{A}_t - \beta D_{pp}^{a_t}(\pi_{\theta_{old}}(\cdot|s_t), \pi_\theta(\cdot|s_t))], \tag{14}
$$

where $\beta$ is the penalized coefficient. These combined advantages lead to considerable performance improvement, which escapes from the dilemma of choosing preferable penalty coefficient. Besides, we use generalized advantage estimates to calculate $\hat{A}_t$. Algorithm 1 shows the complete iteration process of POP3D. Moreover, it possesses the same computing cost and data efficiency as PPO.

---

**Algorithm 1** POP3D

---

1: **Input:** max iterations $L$ , actors $N$, epochs $K$
2: **for** $iteration = 1$ **to** $L$ **do**
3:     **for** $actor = 1$ **to** $N$ **do**
4:         Run policy $\pi_{\theta_{old}}$ for $T$ time steps
5:         Compute advantage estimations $\hat{A}_1, ..., \hat{A}_T$
6:     **end for**
7:     **for** $epoch = 1$ **to** $K$ **do**
8:         Optimized loss objective $f(\theta)$ w.r.t $\theta$ with mini-batch size $M \leq NT$, then update $\theta_{old} \leftarrow \theta$.
9:     **end for**
10: **end for**

---

## 3.7 WORKING MECHANISM OF POP3D

As for the toy example in Section 3.3, $D_{pp}^{RIGHT}(\pi_{\theta_1}(\cdot|s) || \pi_{\theta_2}(\cdot|s)) = D_{pp}^{RIGHT}(\pi_{\theta_1}(\cdot|s) || \pi_{\theta_3}(\cdot|s)) = 0$. Therefore, it can help the agent to focus on the important action. When updating $\theta$ from $\theta_{old}$ as Equation 14, the gradient $f(\theta)$ w.r.t. $\theta$ can be written as

$$\nabla_\theta f(\theta) = \frac{\nabla_\theta \pi_\theta(a_t|s_t)}{\pi_{\theta_{old}}(a_t|s_t)} \hat{A}_t - 2\beta[\pi_\theta(a_t|s_t) - \pi_{\theta_{old}}(a_t|s_t)]\nabla_\theta \pi_\theta(a_t|s_t)$$

$$= \nabla_\theta \pi_\theta(a_t|s_t)[\frac{\hat{A}_t}{\pi_{\theta_{old}}(a_t|s_t)} - 2\beta(\pi_\theta(a_t|s_t) - \pi_{\theta_{old}}(a_t|s_t))] \quad (15)$$

$$= \nabla_\theta \pi_\theta(a_t|s_t)[\frac{\hat{A}_t}{\pi_{\theta_{old}}(a_t|s_t)} - 2\beta\delta(a_t|s_t)]$$

where $\delta(a_t|s_t) := \pi_\theta(a_t|s_t) - \pi_{\theta_{old}}(a_t|s_t)$. Suppose the agent selects $a_t$ at $s_t$ using $\pi_{\theta_{old}}$ and obtains a positive advantage $\hat{A}_t$, if $\pi_\theta(a_t|s_t)$ is larger than $\pi_\theta(a_t|s_t)$, then $2\beta\delta(a_t|s_t)$ will play a damping role to avoid too greedy preference for $a_t$ (i.e. too large probability), which in turn leaves more space for other actions to be explored. Other cases such as negative $\hat{A}_t$ can be analyzed similarly. The hyper-parameter $\beta$ controls the damping force.

In the early stage of learning, $\pi_{\theta_{old}}(a_t|s_t)$ is near $1/K$ (taking $K$ discrete spaces for example) and the magnitude of $\hat{A}_t$ is large, while the damping force is a bit weak. Therefore, the agent learns fast. Then $\beta$ shows a relative stronger force to avoid overshooting for action selection and encourage more exploration. As for the final stage, the policy changes slowly because the learning rate is low, where $\delta(a_t|s_t)$ is small and therefore it converges.

### 3.8 RELATIONSHIP WITH PPO

To conclude this section, we take some time to see why PPO works by taking the above viewpoints into account. When we pour more attention to Equation 9, the ratio $r_t(\theta)$ only involves the probability for given action $a$, which is chosen by policy $\pi$. In other words, all other actions' probabilities except $a$ are not activated, which no longer contribute to back-propagation and allow probability mismatch, which encourage exploration. This procedure behaves similarly to POP3D, which helps the network to learn more easily. Above all, POP3D is designed to conform with the regulations for overcoming above mentioned problems, and in the next section experiments from commonly used benchmarks will evaluate its performance.

## 4 EXPERIMENTS

### 4.1 CONTROLLED EXPERIMENTS SETUP

OpenAI Gym is a well-known simulation environment to test and evaluate various reinforcement algorithms, which is composed of both discrete (Atari) and continuous (Mujoco) domains (Brockman et al., 2016). Most recent deep reinforcement learning methods such as DQN variants (Van Hasselt et al., 2016; Wang et al., 2016; Schaul et al., 2015; Bellemare et al., 2017; Hessel et al., 2018), A3C, ACKTR, PPO are evaluated using only one set of hyper-parameters[5]. Therefore, we evaluate POP3D's performance on 49 Atari games(v4, discrete action space ) and 7 Mujoco (v2, continuous).

Since PPO is a distinguished RL algorithm which defeats various methods such as A3C, A2C ACKTR, we focus on a detailed quantitative comparison with fine-tuned PPO. And we don't consider large scale distributed algorithms Apex-DQN (Horgan et al., 2018) and IMPALA (Espeholt et al., 2018), because we concentrate on comparable and fair evaluation, while the latter is designed to apply with large scale parallelism. Nevertheless, some orthogonal improvements from those methods have the potentials to improve our method further. Furthermore, we include TRPO to acts as a baseline method. Engstrom et al. (2020) carefully study the underlying factor that helps PPO outperform TPRO. To avoid unfair comparisons, we carefully control the settings. In addition, quantitative comparisons between KLD and point probability penalty helps to convince the critical role of the latter, where the former strategy is named fixed KLD in Schulman et al. (2017) and can act as another good baseline in this context, named by **BASELINE** below.

---

[5]DQN variants are evaluated in Atari environment since they are designed to solve problems about discrete action space. However, policy gradient-based algorithms can handle both continuous and discrete problems.

In particular, we retrained one agent for each game with fine-tuned hyper-parameters[6]. To avoid the problems of reproduction about reinforcement algorithms mentioned in Henderson et al. (2018), we take the following measures:

- Use the same training steps and make use of the same amount of game frames(40M for Atari game and 10M for Mujoco).
- Use the same neural network structures, which is the CNN model with one action head and one value head for the Atari game, and a fully-connected model with one value head and one action head which produces the mean and standard deviation of diagonal Gaussian distribution as PPO.
- Initialize parameters using the same strategy as PPO.
- Keep Gym wrappers from Deepmind such as reward clipping and frame stacking unchanged for Atari domain, and enable 30 no-ops at the beginning of each episode.
- Use Adam optimizer (Kingma & Ba, 2014) and decrease $\alpha$ linearly from 1 to 0 for Atari domain as PPO.

To facilitate further comparisons with other approaches, we release the seeds and detailed results[7](across the entire training process for different trials). In addition, we randomly select three seeds from {0, 10, 100, 1000, 10000} for two domains, {10,100,1000} for Atari and {0,10,100} for Mujoco in order to decrease unfavorable subjective bias stated in Henderson et al. (2018).

## 4.2 EVALUATION METRICS

PPO utilizes two score metrics for evaluating agents' performance using various RL algorithms. One is the mean score of the last 100 episodes $Score_{100}$, which measures how high a strategy can hit eventually. Another is the average score across all episodes $Score_{all}$, which evaluates how fast an agent learns. In this paper, we conform to this routine and calculate individual metric by averaging three seeds in the same way.

## 4.3 DISCRETE DOMAIN COMPARISONS

**Hyper-parameters** We search hyper-parameter four times for the penalty coefficient $\beta$ based on four Atari games while keeping other hyper-parameters unchanged as PPO and fix $\beta = 5.0$ to train all Atari games. For BASELINE, we also search hyper-parameter four times on penalty coefficient $\beta$ and choose $\beta = 10.0$. To save space, detailed hyper-parameter setting can be found in Table 6 and 7.

This process is not beneficial for POP3D owing to missing optimization for all hyper-parameters. There are two reasons to make this choice. On the one hand, it's the simplest way to make a relatively fair comparison group such as keeping the same iterations and epochs within one loop to our knowledge. On the other hand, this process imposes low search requirements for time and resources. That's to say, we can draw a conclusion that our method is at least competitive to PPO if it performs better on benchmarks.

**Comparisons** The final score of each game is averaged by three different seeds and the highest is in bold. As Table 1 shows, POP3D outperforms 32 across 49 Atari games given the final score, followed by PPO with 11, BASELINE with 5, and TRPO with 1. Interestingly, for games that POP3D score highest, BASELINE score worse than PPO more often than the other way round, which means that POP3D is not just an approximate version of BASELINE.

For another metric, POP3D wins 20 out of 49 Atari games which matches PPO with 18, followed by BASELINE with 6, and last ranked by TRPO with 5. If we measure the stability of an algorithm by the score variance of different trials, POP3D scores high with good stability across various seeds. And PPO behaves worse in Game Kangaroo and UpNDown. Interestingly, BASELINE shows a large variance for different seeds for several games such as BattleZone, Freeway, Pitfall, and Seaquest. POP3D reveals its better capacity to score high and similar fast learning ability in this domain. The detailed metric for each game is listed in Table 3 and 4.

---

[6]We use OpenAI's PPO and TRPO code: https://github.com/openai/baselines.git
[7]https://drive.google.com/file/d/1c79TqWn74mHXhLjoTWaBKfKaQOsfD2hg/view

## 4.4 CONTINUOUS DOMAIN COMPARISONS

**Hyper-parameters** For PPO, we use the same hyper-parameter configuration as Schulman et al. (2017). Regarding POP3D, we search on two games three times and select 5.0 as the penalty coefficient. More details about hyper-parameters for PPO and POP3D are listed in Table 8. Unlike the Atari domain, we utilize the constant learning rate strategy as Schulman et al. (2017) in the continuous domain instead of the linear decrease strategy.

**Comparison Results** The scores are also averaged on three trials and summarized in Table 1. POP3D occupies 6 out of 7 games on $Score_{100}$. Evaluation metrics of both across different games are illustrated in Table 2

Table 1: **Top**: The number of games "won" by each algorithm for Atari games. **Bottom**: The number of games won by each algorithm for Mujoco games. Each experiment is averaged across three seeds.

| Metric | PPO | POP3D | BASELINE | TRPO |
|---|---|---|---|---|
| $Score_{100}$ | 11 | **32** | 5 | 1 |
| $Score_{all}$ | 18 | **20** | 6 | 5 |

| Metric | PPO | POP3D |
|---|---|---|
| $Score_{100}$ | 1 | **6** |
| $Score_{all}$ | **4** | 3 |

and 5. In summary, both metrics indicate that POP3D is competitive to PPO in the continuous domain.

## 5 CONCLUSION

In this paper, we introduce a new reinforcement learning algorithm called POP3D (Policy Optimization with Penalized Point Probability Distance), which acts as a TRPO variant like PPO. Compared with KLD that is an upper bound for the square of total variance divergence between two distributions, the penalized point probability distance is a symmetric lower bound. Besides, it equivalently expands the optimal solution manifold effectively while encouraging exploration, which is a similar mechanism implicitly possessed by PPO. The proposed method not only possesses several critical improvements from PPO but outperforms with a clear margin on 49 Atari games from the respective of final scores and meets PPO's match as for fast learning ability.

Table 2: Mean final scores (last 100 episodes) of PPO, POP3D on Mujoco games after 10M frames. The results are averaged by three trials.

| Game | PPO | POP3D |
|---|---|---|
| HalfCheetah | 2726.03 | **3184.54** |
| Hopper | **2027.21** | 1452.09 |
| InvertedDblPendulum | 4455.03 | **4907.64** |
| InvertedPendulum | 544.02 | **741.94** |
| Reacher | -5.00 | **-4.29** |
| Swimmer | 111.88 | **112.08** |
| Walker2d | 1112.25 | **3966.01** |

More interestingly, it not only suffers less from the penalty item setting headache along with TRPO, where is arduous to select one fixed value for various environments but outperforms fixed KLD baseline from PPO. In summary, POP3D is highly competitive and an alternative to PPO.

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

# A  SCORE TABLES AND CURVES

Mean scores of various methods for Atari domain are listed in Table 3 and 4.

Table 3: Mean final scores (last 100 episodes) of PPO, POP3D, BASELINE and TRPO on Atari games after 40M frames. The results are averaged on three trials.

| game | POP3D | PPO | BASELINE | TPRO |
|---|---|---|---|---|
| Alien | **1510.80** | 1431.17 | 1311.23 | 1110.40 |
| Amidar | 729.15 | **790.75** | 655.10 | 200.56 |
| Assault | **5400.13** | 4438.82 | 1846.75 | 1363.46 |
| Asterix | **4310.67** | 3483.17 | 3657.67 | 2651.33 |
| Asteroids | **2488.10** | 1605.33 | 1615.37 | 2205.70 |
| Atlantis | **2193605.67** | 2140536.33 | 1515993.33 | 1419104.67 |
| BankHeist | **1212.23** | 1206.67 | 1124.43 | 1125.17 |
| BattleZone | **15466.67** | 14766.67 | 14690.00 | 15123.33 |
| BeamRider | 4549.00 | 2624.19 | **6898.09** | 5073.75 |
| Bowling | 38.99 | **47.27** | 30.48 | 31.24 |
| Boxing | **97.23** | 93.70 | 65.33 | 50.07 |
| Breakout | **458.41** | 281.93 | 67.70 | 40.65 |
| Centipede | 3315.44 | **3565.18** | 3393.93 | 3353.14 |
| Chopper-Command | **6308.33** | 4872.67 | 2676.00 | 2286.67 |
| CrazyClimber | **120247.33** | 105940.00 | 98219.67 | 87522.33 |
| DemonAttack | **61147.33** | 26740.57 | 57476.65 | 21525.08 |
| DoubleDunk | **-7.89** | -11.22 | -8.61 | -10.04 |
| Enduro | 459.85 | **698.46** | 518.41 | 365.95 |
| FishingDerby | **28.99** | 17.72 | -64.27 | -69.64 |
| Freeway | **21.21** | 21.11 | 18.37 | 20.89 |
| Frostbite | **316.87** | 280.30 | 280.30 | 291.77 |
| Gopher | **6207.00** | 1791.00 | 940.87 | 938.27 |
| Gravitar | 557.17 | **753.50** | 449.00 | 495.17 |
| IceHockey | -4.12 | -4.83 | **-3.61** | -4.61 |
| Jamesbond | 527.17 | 488.17 | 685.17 | **901.67** |
| Kangaroo | 3891.67 | **6845.00** | 1850.00 | 1214.67 |
| Krull | 7715.68 | **8329.08** | 7204.95 | 4881.65 |
| KungFuMaster | **33728.00** | 29958.67 | 29843.67 | 26808.00 |
| Montezuma-Revenge | 0.00 | **10.67** | 0.67 | 0.00 |
| MsPacman | 1683.87 | **1981.50** | 1170.70 | 1133.57 |
| NameThisGame | **6065.63** | 5397.47 | 5672.60 | 5604.10 |
| Pitfall | **0.00** | -2.32 | -17.26 | -43.60 |
| Pong | 20.50 | **20.80** | 20.79 | 19.63 |
| PrivateEye | 79.67 | 36.50 | **99.67** | 99.33 |
| Qbert | **15396.67** | 14556.83 | 4114.00 | 3781.58 |
| Riverraid | **8052.23** | 7360.40 | 7722.00 | 6773.67 |
| RoadRunner | **44679.67** | 36289.33 | 43626.33 | 24061.33 |
| Robotank | 4.60 | 14.15 | **24.60** | 24.18 |
| Seaquest | **1807.47** | 1470.60 | 1501.47 | 926.40 |
| SpaceInvaders | **1216.15** | 944.63 | 814.53 | 634.07 |
| StarGunner | **48984.00** | 33862.00 | 47738.00 | 33442.67 |
| Tennis | **-8.32** | -13.74 | -19.13 | -18.40 |
| TimePilot | 3770.33 | 5321.33 | **6278.33** | 5701.00 |
| Tutankham | **241.21** | 177.58 | 135.80 | 136.21 |
| UpNDown | **242701.51** | 153160.66 | 11815.87 | 10949.53 |
| Venture | **36.33** | 0.00 | 4.00 | 0.00 |
| VideoPinball | **37780.70** | 31577.24 | 21438.64 | 25095.20 |
| WizardOfWor | 4704.00 | **4886.67** | 3533.67 | 3103.00 |
| Zaxxon | **9472.00** | 5728.67 | 1179.67 | 4796.67 |

Table 4: All episodes mean scores of PPO, POP3D, BASELINE and TRPO on Atari games after 40M frames. The results are averaged by three trials.

| game | POP3D | PPO | BASELINE | TRPO |
|---|---|---|---|---|
| Alien | **1147.29** | 1115.94 | 851.13 | 841.08 |
| Amidar | 299.55 | **413.46** | 295.91 | 169.12 |
| Assault | 2139.15 | **2168.93** | 1159.50 | 971.78 |
| Asterix | 2004.43 | **2102.10** | 1884.68 | 1342.83 |
| Asteroids | 1652.48 | 1470.46 | 1477.71 | **1760.73** |
| Atlantis | 488134.03 | **596807.27** | 192798.74 | 174394.94 |
| BankHeist | 662.26 | 643.94 | **859.25** | 831.95 |
| BattleZone | 11131.44 | 9387.77 | 11674.30 | **12918.39** |
| BeamRider | 1965.27 | 1460.59 | **3321.25** | 2431.63 |
| Bowling | 37.97 | **39.41** | 33.90 | 30.99 |
| Boxing | **83.12** | 78.61 | 27.92 | 23.07 |
| Breakout | **143.60** | 124.98 | 29.99 | 26.56 |
| Centipede | 3056.81 | **3344.63** | 3042.48 | 3142.22 |
| Chopper-Command | **3269.47** | 3106.14 | 1780.38 | 1595.82 |
| CrazyClimber | **97257.52** | 90169.60 | 69258.31 | 63189.78 |
| DemonAttack | 7611.27 | 7180.43 | **9814.42** | 6204.68 |
| DoubleDunk | **-13.70** | -15.45 | -15.93 | -14.57 |
| Enduro | 107.84 | **321.20** | 92.59 | 140.67 |
| FishingDerby | **-21.00** | -27.51 | -81.90 | -81.97 |
| Freeway | **17.76** | 15.87 | 15.93 | 17.33 |
| Frostbite | **276.47** | 267.73 | 270.42 | 270.57 |
| Gopher | **1556.29** | 1196.20 | 900.74 | 875.93 |
| Gravitar | 413.20 | **509.81** | 342.74 | 317.86 |
| IceHockey | -4.67 | -5.50 | **-4.61** | -5.21 |
| Jamesbond | 358.54 | 394.45 | 380.91 | **519.01** |
| Kangaroo | 1614.63 | **2199.74** | 937.98 | 566.85 |
| Krull | 6538.16 | **7195.24** | 4760.66 | 3861.87 |
| KungFuMaster | 23253.96 | **23283.31** | 19637.58 | 18293.12 |
| Montezuma-Revenge | 0.14 | **0.74** | 0.22 | 0.12 |
| MsPacman | 1214.09 | **1482.77** | 860.63 | 864.84 |
| NameThisGame | **5353.14** | 5199.37 | 4562.32 | 4504.67 |
| Pitfall | **-2.41** | -5.81 | -31.27 | -33.93 |
| Pong | **13.24** | 12.83 | 7.20 | -2.91 |
| PrivateEye | 87.37 | 52.76 | 56.70 | **98.79** |
| Qbert | 5852.10 | **6744.13** | 1760.92 | 1679.03 |
| Riverraid | 5260.89 | **5487.17** | 5220.64 | 4549.22 |
| RoadRunner | **25456.31** | 24688.07 | 20385.91 | 16269.40 |
| Robotank | 3.08 | 8.65 | 13.89 | **14.57** |
| Seaquest | **1487.84** | 1120.15 | 1112.51 | 848.47 |
| SpaceInvaders | **693.26** | 632.17 | 552.50 | 483.48 |
| StarGunner | 14734.11 | 13643.80 | **16288.35** | 13341.23 |
| Tennis | **-19.86** | -21.80 | -21.84 | -21.04 |
| TimePilot | 3396.61 | 4410.87 | **4718.46** | 4544.68 |
| Tutankham | **179.96** | 152.72 | 103.95 | 109.18 |
| UpNDown | 38728.48 | **43208.99** | 5430.22 | 7085.02 |
| Venture | **15.89** | 14.66 | 0.57 | 0.03 |
| VideoPinball | 27346.44 | **27549.55** | 23998.09 | 23705.39 |
| WizardOfWor | 2340.60 | **2743.40** | 2409.94 | 2045.17 |
| Zaxxon | **3739.56** | 1813.90 | 256.78 | 1521.28 |

Table 5: All episodes mean scores of PPO, POP3D on Mujoco games after 10M frames. The results are averaged by three trials.

| game | PPO | POP3D |
|---|---|---|
| HalfCheetah | **3250.22** | 2373.30 |
| Hopper | **1767.14** | 1257.72 |
| InvertedDoublePendulum | **3684.92** | 2561.77 |
| InvertedPendulum | 531.77 | **552.98** |
| Reacher | **-5.94** | -8.05 |
| Swimmer | 94.01 | **108.27** |
| Walker2d | 1770.37 | **2439.54** |

## B  EXPERIMENTS

### B.1  HYPER-PARAMETERS

#### B.1.1  ATARI

PPO's and POP3D's hyper-parameters for Mujoco games are respectively listed in Table 6.

Table 6: **Left**: PPO's hyper-parameters for Atari games. **Right**:POP3D's hyper-parameters for Atari games.

| Hyper-parameter | Value | Hyper-parameter | Value |
|---|---|---|---|
| Horizon (T) | 128 | Horizon (T) | 128 |
| Adam step-size | $2.5 \times 10^{-4} \times \alpha$ | Adam step-size | $2.5 \times 10^{-4} \times \alpha$ |
| Num epochs | 3 | Num epochs | 3 |
| Mini-batch size | 32×8 | Mini-batch size | 32×8 |
| Discount ($\gamma$) | 0.99 | Discount ($\gamma$) | 0.99 |
| GAE parameter ($\lambda$) | 0.95 | GAE parameter ($\lambda$) | 0.95 |
| Number of actors | 8 | Number of actors | 8 |
| Clipping parameter | $0.1 \times \alpha$ | VF coeff. | 1 |
| VF coeff. | 1 | Entropy coeff. | 0.01 |
| Entropy coeff. | 0.01 | KL penalty coeff. | 5.0 |

Table 7: BASELINE's hyper-parameters for Atari games.

| Hyper-parameter | Value |
|---|---|
| Horizon (T) | 128 |
| Adam step-size | $2.5 \times 10^{-4} \times \alpha$ |
| Num epochs | 3 |
| Mini-batch size | 32×8 |
| Discount ($\gamma$) | 0.99 |
| GAE parameter ($\lambda$) | 0.95 |
| Number of actors | 8 |
| VF coeff. | 1 |
| Entropy coeff. | 0.01 |
| KL penalty coeff. | 10.0 |

#### B.1.2  MUJOCO

PPO's and POP3D's hyper-parameters for Mujoco games are respectively listed in Table 8.

Table 8: **Left**: PPO's hyper-parameters for Mujoco games. **Right**:POP3D's hyper-parameters for Mujoco games.

| Hyper-parameter | Value | Hyper-parameter | Value |
|---|---|---|---|
| Horizon (T) | 2048 | Horizon (T) | 2048 |
| Adam step-size | $3 \times 10^{-4}$ | Adam step-size | $3 \times 10^{-4}$ |
| Num epochs | 10 | Num epochs | 10 |
| Mini-batch size | 64 | Mini-batch size | 64 |
| Discount ($\gamma$) | 0.99 | Discount ($\gamma$) | 0.99 |
| GAE parameter ($\lambda$) | 0.95 | GAE parameter ($\lambda$) | 0.95 |
| Clipping parameter | 0.2 | KL penalty coeff. | 5.0 |

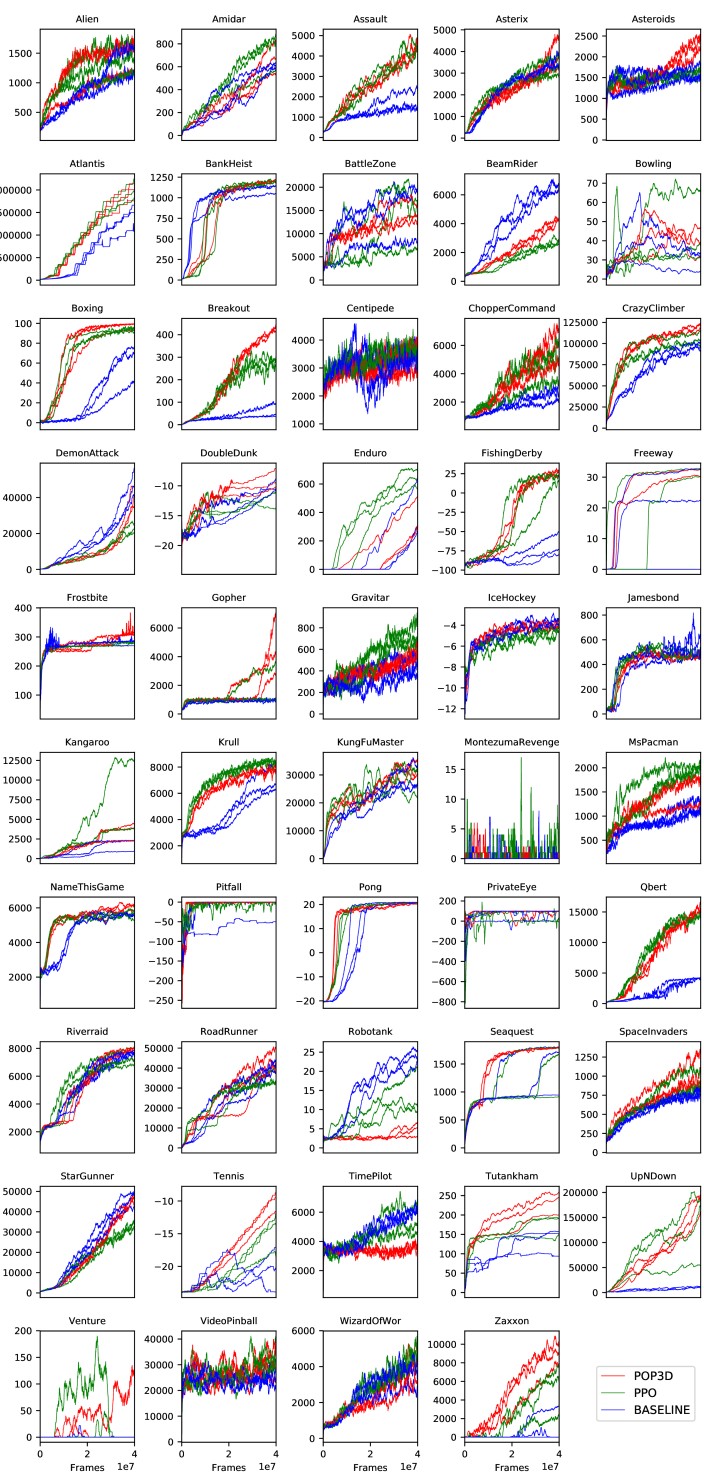

Figure 1: Score curves of three methods on Atari games within 40 million frame steps.

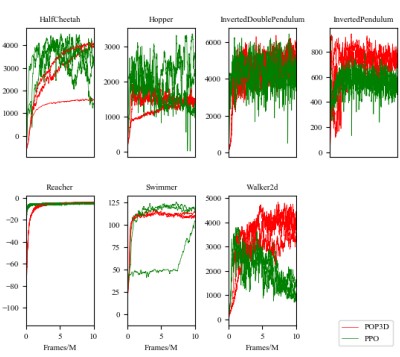

Figure 2: Score curves on 7 Mujoco games within 10 million frame steps.

