# OpenReview forum: "A Strong On-Policy Competitor To PPO"
_ICLR.cc/2021/Conference — Reject_

### Official Review · AnonReviewer4 · 2020-10-28
**Blind Review #4**

**Rating:** 5
**Confidence:** 4

**Review:**

The paper discusses different methods in robust reinforcement learning including TRPO and PPO then propose a variant of TRPO using a new type of regularizer to quantify the difference between two distributions.

Strength:
- The paper is easy to understand and provide informative discussion on related methods.
- The point probability distance is simple but appears to work well.
- The numerical experiments are extensive which include discrete and continuous control tasks.

Weakness:
- There is lack of justification why the proposed algorithm converges.
- Although POP3D appears to be better than PPO and others using the mean final scores metric, it seems to be comparable with PPO when using the all episodes mean scores which raise the concern about the clear improvement of POP3D.

I have the following questions and comments:
Q1. As the point probability distance is bounded, there is an issue with the scale effect where different environment can produce different range of the cumulative reward (the first term in the objective). In this situation, choosing a common \beta that works for most environment appears to be challenging to me? How do you deal with this?

Q2. As from Q1, is there a systematic way to select \beta?

Q3. As the KL divergence is the upper bound of the total variation distance while the point probability distance is the lower bound, POP3D does not inherit from the theoretical result. What is your opinion about this?

C1. Equation (10) appears to be different from the one in the TRPO paper? Are you using a different definition of \eta(.) function, it is better to state its definition for clarity.

C2. It would be better to have figures illustrating the performance of 4 algorithms in some continuous tasks over the number of episodes collected to see their learning process.


Small comments:
- The paper needs to be proofread again for some small typos.
- If the hyper-parameter for POP3D is tuned, it is better if the hyper-parameter for other methods are tuned as well for fair comparison in continuous domain.
- The point probability distance actually depends on the action taken at that iteration. Do you think the formulation needs to be changed to account for that?

Overall, I suggest a weak-reject decision based on the following reason:
- The paper lack theoretical discussion on the convergence of the new method.
- The numerical results do not show significant improvement over existing methods especially PPO as the authors claim.
- The authors claim that the new method removes the need of choosing preferable penalty coefficient while I still believe the parameter \beta needs to be suitably chosen for particular environment.

However, I can adjust my decision once I see the authors’ response.

---

> ### Author Response · Authors · 2020-11-18
> **Adding theoretical discussions, training curves**
>
> **Q1**:  As the point probability distance is bounded, there is an issue with the scale effect where different environment can produce different range of the cumulative reward. There is lack of justification why the proposed algorithm converges.
>
> **A1**:
> Thanks for this question. We are glad to see that it's raised. We didn't use the  cumulative reward (the first term in the objective) in Eq.14, instead we use generalized advantage (Eq.6, the same as PPO.) Moreover, the reward in each game has been normalized as DQN[2] and PPO. Besides, the running mean and std of reward is utilized to control the first item as PPO. Thus, the first item in fact is also bounded. And we have provided the code to make it clear.
> The second one $D_{pp}$ is a type of regulation for policy updates. Note that $D_{pp}$ lies in [0,1] and most cases $D_{pp}$ < 0.1.  Therefore, it's easy to set the first item as the dominant item (about 1.0~3.0). During the learning process, the return is improved. When the return saturates, the policy also becomes stable, meaning the second item also contributes less to the total loss. In this way, it converges. We provided all the training logs for the experiments in the supplementary, which can empirically prove its convergence. For example, the experiment of Alien (one Atari game) shows the loss of the second item decreases from -0.054 to -9.2e-06.
>
> **Q2**: Is there a systematic way to select $\beta$?
>
> **A2**: Excuse for the confusion, and we've revised it more accurately. We believe A1 has clarified something about the bound of each loss item. We don't mean that our method removes the need of choosing the penalty coefficient at all, rather it eases the pain for the selection. TRPO [1] evolves from its penalized version (Section 4) where the penalized coefficient $C$ is hard to calibrate in practice. Therefore, they used a trust-region approach. PPO also mentioned this issue: "TRPO uses a hard constraint rather than a penalty because it is hard to choose a single value of $\beta$ that performs well across different problems—or even within a single problem".
> Although ours is also a penalized method, it is easier to set the coefficient $beta$ with common practice: i.e.  using a small subset of games to tune the hyper-parameters. Note that, we only tune the $beta$  in 3 games to set the $beta$ for all 49 Atari games because it is newly introduced. We keep other hyper-parameters unchanged as PPO although it maybe doesn't fit our method well. PPO also has an extra $\epsilon$ to be tuned.  We study the value of regularized item and others in the loss. The magnitude  of former ($\beta$=5) is about 0.05 $\times$ of  the latter.  Note that we should use the reward normalization and advantage for a new environment.
>
> **Q3**: As the KL divergence is the upper bound of the total variation distance while yours is the lower, POP3D does not inherit from the theoretical result. What is your opinion about this?
>
> **A3**: Thanks for this question. This is actually the value of our work.   As a special case, it can be easily proved that for binary distribution, $D_{TV}^2(p||q) = D_{pp}(p||q)$. Different from the upper bound approach, the new lower bound of POP3D no longer requires exact distribution matching  between new and old policy. Instead, it pays attentions to the most important action at a given state, which helps the policy network learn better and more easily. This approach directly encourages more explorations. This role can be better understood in section 3.3 and 3.7 (revised version).
>
> **Q4**: It is better if the hyper-parameter for other methods are tuned as well for fair comparison in the continuous domain.
>
> **A4**: Yes. Note that we only For BASELINE, we have done it in Section 4.3. For PPO, we use their tuned hyper-parameters to report the result.
>
> **Q5**: Equation (10) appears to be different from the one in the TRPO paper.
>
> **A5**: We have clarified it in the rebuttal version. This is because we cited the ICML version [1] instead of the arxiv one. In their published version, $\eta$ means the loss instead of a reward.
>
> **Q6**: It would be better to have figures illustrating the performance of  the algorithm to see their learning process.
>
> **A6**:  We have updated the score curves to have a better view of the learning process (including both Atari and Mojuco).
>
> **Q7**: The point probability distance actually depends on the action taken at that iteration.
>
> **A7**: We have revised it in the rebuttal version using $D^a_{pp}$.
>
> We carefully  proofread our paper again in the revised version.
>
> **References**
> 1. Schulman et.al, Trust region policy optimization, ICML 2015
> 2. Mnih  et.al Human-level control through deep reinforcement learning Nature

---

### Official Review · AnonReviewer1 · 2020-10-28
**A simple surrogate objective with almost no insight**

**Rating:** 5
**Confidence:** 4

**Review:**

The authors replace the divergence-based constraint in trust region policy optimization model with an alternate distance measure, which is added to the objective function with a multiplier (beta).  In fact, the parameter beta plays a role that is similar to a Lagrange multiplier, if the new distance measure is introduced as a constraint. The authors explain the shortcomings of KL-divergence and the solutions obtained with other methods but they do not provide a sufficient discussion how and why their simple approach overcomes those concerns. For instance, why would the new measure encourage exploration and what is the effect of large beta value on this?

---

> ### Author Response · Authors · 2020-11-18
> **More discussions about how and why our simple approach overcomes those concerns.**
>
> Our work shows that  lower bounding the square of the total variance divergence is meaningful and achieves comparable results as PPO. To our best knowledge, our work is the first one that steps out of the upper bounding framework and achieves comparable to PPO.   The upper bounding framework  is popular and even dominates since TRPO.  Our method is simple and efficient.   During the rebuttal stage, we strengthen it from the theoretical aspect.
> Therefore, we believe it will bring in many insights  to the RL community.
>
> **Q1**: Why would the new measure encourage exploration?
>
> **A1**: We further discuss why POP3D  overcomes those concerns in section 3.7 in the revised version. We hope we can help you better understand its working mechanism. $2\beta [\pi_{\theta}(a_t|s_t)-\pi_{\theta_{old}}(a_t|s_t)]$ will play a damping role to avoid too greedy preference for $a_t$ (i.e. too large probability), which in turn leaves more space for other actions to be explored.
> We believe the example used in Section 3.3 can help it better understood.
>
> **Q2**: What is the effect of large beta value?
>
> **A2** :The role of $\beta$ is further analyzed in Section3.7.
> The gradient the surrogate objective w.r.t $\theta$ is
> $\nabla_\theta f(\theta)=\nabla_{\theta}\pi_\theta(a_t|s_t)[\frac{\hat{A_t}}{\pi_{\theta_{old}}(a_t|s_t)}-2\beta (\pi_{\theta}(a_t|s_t)-\pi_{\theta_{old}}(a_t|s_t))]$.
>  The hyper-parameter $\beta$ plays a role of  avoid overshooting for action selection and encourage more exploration.
> Large $\beta$ will dominate the gradient and make the agent learn slowly or even fail to learn.

---

### Official Review · AnonReviewer2 · 2020-10-29
**This paper introduces POP3D, an on-policy policy gradient algorithm that is a variant of TRPO and PPO. Overall, this was an interesting work. I believe that the idea of lower bounding the square of the total variance divergence is worth being investigated. The experimental results seem promising, though I do have questions regarding the statistical significance of the results. I also believe that this paper might be improved by clarifying a few of the authors' key mathematical arguments.**

**Rating:** 5
**Confidence:** 4

**Review:**

This paper introduces POP3D, an on-policy policy gradient algorithm that is a variant of TRPO and PPO. While TRPO uses a particular penalty function to keep the policy from being updated too aggressively, POP3D uses an alternative objective function that lower bounds the square of the total variance divergence between two policy distributions. The authors argue that this alternative formulation results in an algorithm that is sample-efficient, like PPO, but that is more effective at keeping policy updates from overshooting. The authors also argue that this new formulation helps users to avoid the arguably challenging process of selecting penalty constants, as required (for instance) by TRPO.

Overall, this was an interesting paper. I believe that the idea of lower bounding the square of the total variance divergence is worth being investigated. The experimental results also seem promising, showing that the method may outperform POP and TRPO on a wide range of games. I do have a few questions about the statistical significance of the results, though: some of the score differences may not be significant, the results might be based on too few trials, and the authors only presented mean performances, but not standard deviation or standard error information. I liked the idea that the proposed lower bound solves possible issues arising from using the KL divergence, which is a measure that is not symmetric. On the other hand, after presenting the surrogate objective function's mathematical definition, the authors argued that this new formulation results in better exploration and that it "expands the manifold [of] solutions". I had a hard time following these arguments.

Finally, in my opinion, writing could be improved. A few sentences are hard to read or appear to be presenting incomplete thoughts. For instance:

- "Hessian free strategy: Fisher vector product is utilized to cut down the computing burden." (page 1)
- "(...) the action is taken approximately strongly corrected with the highest probability value." (page 4)
- "Therefore, even if theta outputs theta_old the same high probability for the right action, it's still penalized owing to probabilities mismatch for other uncritical actions." (page 4)
- "From the perspective of the manifold, if the optimal parameters constitute a solution manifold." (page 4)

I believe that this is an interesting work, though possibly still showing only preliminary results. I also think that the paper might be improved by clarifying a few of the authors' key mathematical arguments. As it is, and based on the thoughts presented above (and on the questions below), I would say that more work would benefit the paper and would argue for a weak reject.

I have a few questions and comments for the authors:

1) the authors state the "[POP3D] dives into the mechanism of PPO's improvement over TRPO by the perspective of solution manifold". Could you please clarify? What is the solution manifold perspective, and how is it used more effectively by POP3D, compared to previous algorithms? At the end of Section 3.3, you present an argument that I found hard to follow: "From the perspective of the manifold, if the optimal parameters constitute a solution manifold.". Later on, you also talked about "[expanding] the solution manifold at least one dimension such as curves to surfaces". I believe that these ideas may be central to the paper. Still, I had difficulty following the arguments in this section.

2) a quick note on notation: \hat{A}_t (Eq6) looks like a random variable denoting the action taken in time t. Perhaps the LHS of this equation is missing parameters specifying a (s,a) pair? In particular, note that the advantage function in Eq6 involves computing delta^V_{t+l}, a TD error for a specific experience; the current notation does not make it clear which experiences are being analyzed. Compare, for example, the definition of advantage used in Eq6 and the one used in Eq3.

3) still regarding notation, I wonder if you could clarify which of s, a, s_t, a_t, are random variables and which ones are realizations of the corresponding random variables. In Eq6, for example, why do you write A(s_t, a) instead of A(s_t, a_t)?

4) in Eq10, please formally define D_{TV}.

5) immediately after Eq11, the authors argue that because the KL divergence is not symmetric, the choice of whether to use D_KL(p1, p2) or D_KL(p2, p1) might affect the algorithm's behavior, and that (for this reason) it might not be the best choice for a divergence measure. Could you please discuss the possible advantages of simply using the symmetrized version of this divergence (https://en.wikipedia.org/wiki/Kullback%E2%80%93Leibler_divergence#Symmetrised_divergence)?

6) in Eq12, where is the action "a", on the RHS, coming from? The inputs to D_pp are two distributions, but its definition seems to consider the difference between two particular probabilities. Could you please clarify the notation being used here?

7) regarding your experiments, I wonder if you could discuss whether the score differences observed in Table 1 are statistically significant.

8) your method does seem to perform better in some particular games, but it does not do so well in other games. A discussion/interpretation of these results would be nice: what are, for instance, the properties of the games that are hard for POP3D?

9) regarding Table 3, could you please discuss whether you can draw strong conclusions regarding the performance difference between PPO and POP3D when analyzing just 3 trials? Also, do you have standard deviation information associated with the averages that you could present?

10) one key point that the authors make is that POP3D encourages exploration by "expanding the optimal solution manifold". More efficient exploration is indeed advantageous, but I did not follow the argument being made here. Could you please clarify why POP3D results in better exploration?

---

> ### Author Response · Authors · 2020-11-18
> **clarifying a few  key mathematical arguments and statistical  results**
>
> To make fair comparisons, we carefully control the experiment settings and **exactly follow the evolution protocols used in PPO**. And we provide the learning curves for all experiments to see the variance. We also provide the code and training logs to make our work reproducible.
>
> **Q1**: Clarify  the mechanism of PPO's improvement over TRPO by the perspective of solution manifold". And how is it used more effectively by POP3D, compared to previous algorithms?
>
> **A1**：Indeed, we have carefully revised the last paragraph of Section 3.3 with a clearer explanation. From the perspective of the manifold, learning with POP3D's loose constraint is akin to mapping a point to a surface (soft and easy) other than KLD's strict constraint that forces mapping it onto a curve (more accurate by difficult).
>
> **Q2**: Clarify notation in  (Eq6). Compare, for example, the definition of advantage used in Eq6 and the one used in Eq3.
>
> **A2**: We follow [3] for the notation $\hat{A}t$ in (Eq6), and this is clarifed to make it self-contained (revised version). All methods in this paper use Eq.6 to estimate advantage. As stated in [3], Eq. 6 is a generalized (the G of GAE) and weighted form to Eq3. When $\lambda=0$, Eq.6 is equivalient to Eq.3.
>
> **Q3**: Notation which of $s$, $a$, $s_t$, $a_t$, are random variables and which ones are realizations of the corresponding random variables. In Eq.6, for example, why do you write $A(s_t, a)$ instead of $A(s_t, a_t)$?
>
> **A3**: $s$,$a$ in Eq.3 are random varables. $s_t$, $a_t$ are observations.  $A(s_t, a)$ indeed should be $A(s_t, a_t)$. We have revised these notations to make it clear.
>
> **Q4**: Formally define $D_{TV}$ in Eq10.
>
> **A4**: Revised as in Equation 13 for the rebuttal version.
>
> **Q5**: Discuss the possible advantages of simply using the symmetrized version of this divergence.
>
> **A5**: We have tried JS divergence (Jensen-Shannon Divergence) and D_KL(p1, p2) + D_KL(p2, p1) previously. However, they win PPO with a  small advantage. Although JSD is symmetric and bounded, it  still suffers from limited exploration. Motivated by the important observation that  clipped advantage is quite critical for PPO,  we realize the importance of exploration. The surrogate objective of PPO is regarded pessimistic, which means that prohibiting greedy policy matters. This motivates us to find more effective regularization that can encourage exploration.
>
> **Q6**:  Clarify the notation for Eq12 (submission version)?
>
>
> **A6**: The action $a$ is the one that an agent takes (following its policy) at state $s$. We have clarified the notation in Equation 13 of the revised version.
>
> **Q7**: Whether the score differences observed in Table 1(submission) are statistically significant.
>
> **A7**: We exactly follow the same evaluation protocol as PPO: all experiments are reported across 3 trials and evaluating the performances using the same metrics within 40M frames for Gym Atari.  We have updated the accumulated reward curves to see the variance and learning process (see Figure 1).  We also report the two metrics as PPO. (1) average reward per episode over the entire training period (which favors fast learning), and (2) average reward per episode over last 100 episodes of training (which favors final performance). And Table 1 (revised version) shows our POP3D outperforms the baselines. Note that PPO utilizes the same strategy and proof to defeat other counterparts.
>
> **Q8**: A discussion/interpretation of these results would be nice: what are, for instance, the properties of the games that are hard for POP3D?
>
> **A8**: Thanks for the suggestion. Based on the averaged performance, it seems that POP3D behaves better when the game is harder (like Gopher, Zaxxon and Pitfall) where more explorations are required.
>
> **Q9**: Regarding Table 3 (submission), strong conclusions regarding the performance difference between PPO and POP3D when analyzing just 3 trials? Also, do you have standard deviation information associated with the averages that you could present?
>
> **A9**. We use exactly the same metrics and settings as [2] to make fair comparisons. PPO [2] reports their results using 3 trials. We have updated the accumulated reward curves to see the variance and learning process (see Figure 2).  Moreover, we also report the two metrics as PPO. Table 2 in revised version shows our POP3D is comparable (used in Section4.4).
>
> **Q10**. Why POP3D results in better exploration?
>
> **A10** Please see the toy example in section 3.3.
>
> We carefully proofread our paper  in the revised version.
>
> **References**
>
> 1. Schulman et.al, Trust Region Policy Optimization, ICML 2015
> 2. Schulman et.al, Proximal Policy Optimization Algorithms, Arxiv 2017
> 3. Schulman et.al HIGH-DIMENSIONAL CONTINUOUS CONTROL USING GENERALIZED ADVANTAGE ESTIMATION, ICLR 2016

---

### Author Response · Authors · 2020-11-19
**A summary of updates in PDF of POP3D**

We thank everyone  for the dedicated reviews.
Here is  a summary of the PDF.
1. A section 3.7 is added to theoretically analyze the working mechanisms of our method.
2. Learning curves for Atari and Mujoco are added to better see the learning process and variance during the learning.
3. Improve section 3.3 to make our motivations better understood, as well as the role of POP3D.

---

### Author Response · Authors · 2020-11-19
**Common Responses**

We thank the reviewers for their meaningful and valuable comments, which help to improve the quality of our work.

Our work shows that  lower bounding the square of the total variance divergence is meaningful and achieves comparable results as PPO. To our best knowledge, our work is the first one that steps out of the upper bounding framework and achieves comparable to PPO.   The upper bounding framework  is popular and even dominates since TRPO.
Our method is quite simple and efficient.

During the rebuttal, we strengthen it from the theoretical aspect (section 3.7) and further analyze the mechanism of our method.

TRPO and PPO are two of the most influential methods in RL community. They both tried the penalized approach but don't succeed. In contrast, our simple approach does.

Moreover, we carefully control the experiment to make fair comparison. And We have provided the code and training logs to make work reproducible and convincing.

Therefore, we believe it will bring in some insights  to the RL community. We sincerely hope that the reviewers can reevaluate the value of our paper. We are looking forward to your replies.

---

### Decision · Program_Chairs · 2021-01-07
**Final Decision**

**Decision:**

Reject

**Comment:**

While the paper contains some interesting ideas, the reviewers felt that overall the paper is not theoretical well supported, and likewise the experiments are not fully convincing. Even after the rebuttal, these concerns still persist.